Rapid restructurization of conformationally-distinct alpha-synuclein amyloid fibrils at an elevated temperature

Ziaunys Mantas mantas.ziaunys@gmail.com
Sakalauskas Andrius
http://orcid.org/0000-0002-4826-9856 Mikalauskaite Kamile
http://orcid.org/0000-0002-1829-5455 Smirnovas Vytautas
Institute of Biotechnology, Life Sciences Center, Vilnius University , Vilnius , Lithuania
Uversky Vladimir
Electronic publication date: 2022 Sep 30
Publication date: 2022
Volume: 10
Electronic Location ID: e14137
Received 2022 Jun 2; Accepted 2022 Sep 6
Copyright: © 2022 Ziaunys et al.
Copyright year: 2022
Copyright holder: Ziaunys et al.
License: This is an open access article distributed under the terms of the Creative Commons Attribution License, which permits unrestricted use, distribution, reproduction and adaptation in any medium and for any purpose provided that it is properly attributed. For attribution, the original author(s), title, publication source (PeerJ) and either DOI or URL of the article must be cited.
License URL: https://creativecommons.org/licenses/by/4.0/

Keywords: Amyloid, Alpha-synuclein, Protein aggregation, Protein fibrils, Fibril structure

Funding: The authors received no funding for this work.

==============================
Protein aggregation in the form of amyloid fibrils is linked with the onset and progression of more than 30 amyloidoses, including multiple neurodegenerative disorders, such as Alzheimer’s or Parkinson’s disease. Despite countless studies and years of research, the process of such aggregate formation is still not fully understood. One peculiar aspect of amyloids is that they appear to be capable of undergoing structural rearrangements even after the fibrils have already formed. Such a phenomenon was reported to occur in the case of alpha-synuclein and amyloid beta aggregates after a long period of incubation. In this work, we examine whether incubation at an elevated temperature can induce the restructurization of four different conformation alpha-synuclein amyloid fibrils. We show that this structural alteration occurs in a relatively brief time period, when the aggregates are incubated at 60 °C. Additionally, it appears that during this process multiple conformationally-distinct alpha-synuclein fibrils all shift towards an identical secondary structure.

Introduction

Protein aggregation into highly-structured amyloid fibrils is linked with the onset and progression of over 30 amyloidoses, including neurodegenerative Alzheimer’s or Parkinson’s diseases (Knowles, Vendruscolo & Dobson, 2014; Chiti & Dobson, 2017). Currently, the process of fibril formation is still not completely understood (Lee & Terentjev, 2017; Chaudhuri et al., 2019), which, in turn, has resulted in very few approved drugs or treatment modalities (Mehta et al., 2017; Doig et al., 2017; Maurer et al., 2018; Cummings et al., 2020). It is projected that the already high number of patients afflicted with these disorders will continue to rise in the upcoming years (Brookmeyer, Gray & Kawas, 1998; Arthur et al., 2016). In order to prevent this and discover potential cures, a better understanding of both the formation pathways (Brännström et al., 2018), as well as the resulting aggregates (Fändrich et al., 2018) is necessary.

Protein misfolding and association into fibrillar structures proceeds through multiple stages. The first step in amyloid aggregation is primary nucleation—a process during which native protein molecules misfold and assemble into a stable, β-sheet-rich nucleus (Buell et al., 2014; Meisl et al., 2014). Once a stable nucleus is formed, it can then incorporate other, homologous protein molecules into its structure and elongate into a protofibrillar aggregate. The process of elongation occurs at the ends of the fibrillar structure, which act as both a catalyst and template for the conversion of native protein/peptide molecules (Milto, Botyriute & Smirnovas, 2013; Gurry & Stultz, 2014; Rodriguez et al., 2018). When amyloid fibrils reach a critical length, they can experience fragmentation—a process during which the aggregate breaks into two shorter fibrils, each with its own aggregation-catalyzing ends. The incidence of fragmentation depends on both the structural stability of the aggregate, as well as the environmental conditions (Nicoud et al., 2015). Another process, which occurs once amyloid fibrils form, is surface-mediated nucleation (also referred to as secondary nucleation) (Foderà et al., 2008; Gaspar et al., 2017; Törnquist et al., 2018). During this step, the surface of aggregates acts as a catalyst for the formation of new amyloid nuclei. Unlike elongation at fibril ends, this process does not template the aggregate’s structure onto the nuclei, but only increase their probability of formation (Sneideris, Milto & Smirnovas, 2015).

It is usually considered that once the aggregation reaction uses up all or most of the available monomers, then the amyloid formation process is complete, with only certain other events occurring, such as fibril lateral association (Fujiwara, Matsumoto & Yonezawa, 2003) or large cluster formation (Manno et al., 2006). However, this may not be the case. It was shown that during a long incubation time, alpha-synuclein (Sidhu et al., 2017) and amyloid beta fibrils (Ma et al., 2013) experienced structural alterations. It was hypothesized that this type of aggregate “maturation” may also be required for fibrils to obtain a higher level of infectivity (Yamaguchi et al., 2005). If this restructurization occurs for other amyloids as well, then it may also have an impact on in vitro experiments, making incubation time (Morel et al., 2010) a crucial factor. Even if the exact same reaction solution and protocol is used, the aggregates may undergo structural rearrangement after their formation (Mahdavimehr, Katebi & Meratan, 2018). If the assay employs a set timeframe of fibrillization, then uninhibited aggregation would yield fibers sooner than in the inhibitor reaction solution and they would incubate for a relatively longer time. This could result in aggregates with seemingly different structural aspects (Pellarin et al., 2010) and skew the experimental data, as well as conclusions of the study.

In this work, we generated alpha-synuclein fibrils under previously reported experimental conditions, which yield multiple structurally-distinct aggregates from an identical protein solution (Toleikis et al., 2021). The resulting four distinct fibril solutions were then incubated at 60 °C and their structural alterations were observed using CD, infrared and UV/Vis spectroscopy, atomic force microscopy, as well as ThT-binding assays (Hoyer et al., 2002; Ziaunys, Sakalauskas & Smirnovas, 2020). We show that all four different types of fibrils rapidly converge into aggregates with similar secondary structures and that this change occurs in the matter of hours.

Materials and Methods

Initial fibril preparation

Alpha-synuclein (α-syn) was purified as described previously (Šneideris et al., 2015), lyophilized, and stored at −20 °C prior to use. Further fibril preparation procedures were done under similar conditions as described previously (Toleikis et al., 2021). The α-syn powder was dissolved in a phosphate-buffered saline (PBS, pH 7.4) and filtered through a 0.22 µm syringe filter, after which the protein concentration was determined using a Shimadzu UV-1800 spectrophotometer (ε280 = 5,960 M−1cm−1). The protein solution was then combined with a 10 mM thioflavin-T (ThT) solution and PBS to result in a final protein concentration of 70 and 100 µM ThT. The solution was then distributed to 96-well half-area non-binding plates (cat. No 3881, Fisher Scientific, Hampton, NH, USA) (each well contained 100 µL of the solution and one 3 mm diameter glass-bead (cat. No 104015, Merck, Kenilworth, NJ, USA) and sealed with Nunc sealing-tape. The 96-well plates were incubated at 37 °C in a ClarioStar Plus plate reader, under constant 600 RPM agitation for 48 h, with measurements taken every 5 min (excitation wavelength was 440 nm, emission – 480 nm). For aggregation monitoring under 60 °C, all sample preparation and incubation procedures, apart from the set temperature, were identical. After aggregation, the plates were cooled down to room temperature prior to further examination. Aggregation lag time and apparent rate constants were determined as described previously (Ziaunys et al., 2021a).

Excitation-emission matrices (EEM)

The procedure of EEM acquisition was based on a previous study on alpha-synuclein aggregation (Ziaunys et al., 2021a). In short, 96-well plates containing the fibril samples were placed in a ClarioStar Plus platereader and incubated at 25 °C for 5 min. Bound-ThT excitation-emission matrices (EEM) were obtained by first scanning the sample fluorescence emission intensity at a fixed wavelength (485 nm), using a range of excitation wavelengths (430–460 nm). Afterwards, the intensity was scanned under a range of emission wavelengths (470–500 nm), using a fixed excitation wavelength (445 nm). The data was combined into an EEM by using ClarioStar MARS 3D spectrum function. EEM “centers of mass” were calculated as described previously (Ziaunys & Smirnovas, 2019). In short, the center of mass was calculated for the top 10% intensity values from each EEM. All the maximum position values were then plotted in a single graph for comparison. Since alpha-synuclein and other amyloid fibril ThT-binding characteristics can be related to their structure/morphology (Sidhu et al., 2018; Ziaunys et al., 2021a; Ziaunys et al., 2021b), eight out of the 96 samples with distinct EEM maximum positions were selected for further replication and analysis. This selection procedure was done in order to improve the likelihood of obtaining different types of alpha-synuclein fibrils, without the necessity to analyze the entirety of the large number of samples.

Fibril reseeding

In order to generate a higher quantity of aggregates, the eight selected samples (100 µL) were each combined with 400 µL of the initial reaction solution, containing 70 µM α-syn monomers. The resulting solutions were placed in 1.5 mL test-tubes (each containing two 3 mm glass beads) and incubated at 37 °C with constants 600 RPM agitation for 24 h in a Digital Heating Shaking Dry bath (Fisher Scientific, Hampton, NH, USA). The resulting aggregate solutions were then combined with 2 mL of the initial reaction solution, divided into 1.5 mL test-tubes (500 µL each) and incubated identically for another 24 h. The resulting 2.5 mL solutions were combined with 2.5 mL of initial reaction solutions and incubated as described previously, resulting in 5 mL volume α-syn aggregate solutions, which were then cooled down to room temperature.

Each fibril solution was then distributed to 1.5 mL test-tubes (1 mL each) and centrifuged at 10,000×g for 15 min, after which the supernatant (900 µL) was removed and replaced with an identical volume of the initial buffer solution, which did not contain α-syn monomers. These centrifugation and fibril pellet resuspension steps were repeated three times, in order to remove non-aggregated protein monomers. This was done in case the aggregation reactions at 37 °C did not use up all available monomers (the resulting samples may also contain a fraction of monomers due to an equilibrium between native and aggregated proteins). Afterwards the samples were combined to a final volume of 5 mL. All further experimental procedures were performed using only these replicated and resuspended samples.

Fibril incubation

Each 5 mL fibril sample was vigorously agitated for 10 s and then distributed to sets of three 1.5 mL test-tubes (1.5 mL solution volume, each test-tube contained two 3 mm glass beads). The first set was kept at room temperature, while the other two sets were incubated at 60 °C under constant 600 RPM agitation for either 24 or 48 h. During the incubation procedures, aliquots of each sample were taken for further analysis by Fourier-transform infrared (FTIR) and circular dichroism (CD) spectroscopies, atomic force microscopy (AFM) and UV/Vis spectroscopy.

To measure the changes in bound-ThT EEM positions, the selected samples were placed into 96-well half-area non-binding plates (100 µL volume, one 3 mm glass bead in each well) and incubated in a ClarioStar Plus plate reader at 60 °C under constant 600 RPM agitation. EEMs were scanned every hour. Both the scanning and analysis procedure was identical to the one described in the previous method section.

Fourier-transform infrared (FTIR) spectroscopy

For FTIR measurements, 500 µL of each aggregate sample was centrifuged at 10,000 × g for 15 min, after which the supernatant was removed and replaced with 500 µL D2O (containing 400 mM NaCl in order to improve fibril sedimentation (Mikalauskaite et al., 2020)). The centrifugation and resuspension procedure was repeated four times. After the final step, the fibril pellet was resuspended into 50 µL D2O and mixed vigorously for 10 s. FTIR spectra were acquired as described previously (Mikalauskaite, Ziaunys & Smirnovas, 2022), which were then baseline corrected and normalized to the same band area in a range between 1,700 and 1,595 cm−1. All data processing was done using GRAMS software.

Circular dichroism (CD) spectroscopy

For CD spectroscopy measurements, aliquots of each fibril sample (50 µL) were taken every hour for the first 10 incubation hours, then after 24 and 48 h. The samples were cooled down to 25 °C and placed in a 0.1 mm pathlength cuvette. The CD spectra were measured between 200 and 250 nm, using a Jasco J-815 spectropolarimeter. For each sample, three spectra were recorded and averaged, after which a PBS solution spectrum was subtracted from each sample spectrum. The data was then analyzed using BeStSel Protein Circular Dichroism (Micsonai et al., 2015) spectra analysis software, in order to determine the content of secondary structures.

Atomic force microscopy (AFM)

AFM sample preparations and measurements were performed as described previously (Ziaunys et al., 2021a). Before sample deposition, freshly cleaved mica surface was modified with (3-aminopropyl) triethoxysilane (APTES). 0.5% (% v/v) APTES solution (30 µL) was spread on the surface of the mica, incubated at room temperature for 5 min, gently washed with 2 mL of H2O and dried using airflow. 30 µL aliquots of each sample were placed on APTES-modified mica and left to adsorb for 60 s. The mica were then gently washed with 3 mL of H2O and dried using airflow. AFM images were acquired using a Dimension Icon (Bruker) atomic force microscope and analyzed using Gwyddion 2.5.5 software as described previously (Ziaunys et al., 2021a).

UV/Vis spectroscopy

Each fibril sample (500 µL) was supplemented with 5 µL 10 mM ThT and vigorously agitated for 10 s. The samples were then placed in a 3 mm pathlength cuvette and sample absorbance spectra were scanned in the 200–600 nm range using a Shimadzu UV-1800 spectrophotometer (three repeats were scanned for each sample and averaged). After this, 200 µL of each sample was centrifuged at 10,000×g for 15 min and 100 µL of the supernatant was carefully removed. The supernatant absorbance spectra were scanned as described previously. In order to determine the absorbance spectra of bound-ThT molecules, the supernatant spectra were subtracted from the fibril-ThT spectra and baseline corrected between 300 and 550 nm using Origin 2018 software baseline subtraction function with three anchor points at each side of the peak.

Results

In order to test the effect that an elevated temperature has on alpha-synuclein (α-syn) fibril structure, different conformation aggregates were first generated. Since α-syn is capable of spontaneously forming structurally distinct fibrils under identical experimental conditions (Toleikis et al., 2021; Ziaunys et al., 2021a) at random, a large number of samples (n = 96) was aggregated for an initial assessment. Potential conformationally-distinct fibrils were then identified by scanning their bound-ThT excitation-emission matrices (EEM) and selecting eight samples from the set, based on their unique EEM positions (Fig. 1A) (Ziaunys et al., 2021a). This was done in order to increase the likelihood of obtaining different structure samples, without the necessity to analyze the entire set. The selected samples were then replicated as described in the Materials and Methods section in order to generate a higher quantity of aggregates. To determine which of the samples contained significantly different secondary structures, an aliquot from each was scanned using Fourier-transform infrared spectroscopy (FTIR).

Figure 1 Excitation-emission matrix (EEM) positions of fibril-bound ThT fluorescence (A) and selected fibril sample FTIR spectra (B) and their second derivatives (C).

EEM maximum positions were determined as described in the Materials and Methods section (96 samples) after sample aggregation. Red color-coded circles marked with Roman numerals represent samples chosen for further analysis.

Five of the samples contained aggregates with similar secondary structures (Figs. 1B, 1C), with the FTIR spectrum main maximum at 1,627 cm−1 and a shoulder at 1,636 cm−1 (same position minima in the second derivatives, which are positions associated with hydrogen bonds in the beta-sheet structure (Barth, 2007)). They also had a minimum in the second derivative at 1,666 cm−1, which is related to turn/loop motifs. Due to their similarities, these five samples were regarded as Type 1 fibrils. The VI sample shared some similarities to Type 1 aggregates in the turn/loop motif position (1,666 cm−1), however, the main maximum was shifted to 1,624 cm−1, suggesting a different type of hydrogen bonding in the beta-sheet structure. Since this sample FTIR spectra was different from all other seven and had a main maximum position between other two group spectra positions, it was regarded as consisting of Type 2 fibrils. The other two remaining samples (VII and VIII) shared a similar main maximum position at 1,623–1,624 cm−1, with significant variation in the turn/loop motif regions (minima at 1,662 and 1,673 cm−1 for Type 3, 1,668 cm−1 for Type 4). These four replicated fibril types were then used in all further experimental procedures.

Before incubating the selected fibrils at an elevated temperature, the aggregates were first centrifuged and resuspended into their initial buffer solution, in case they contained a significant concentration of non-aggregated protein. This was done to avoid any possible additional aggregation occurring during the incubation procedure. The resuspended fibrils were then incubated at 60 °C with constant agitation for 24 or 48 h as described in the Materials and Methods section. After this step, a portion of each sample was used to scan their FTIR spectra.

In the case of Type 1 fibrils (Figs. 2A, 2E) after 24 h of incubation, there was a shift of the FTIR spectrum main maximum position towards 1,625–1,626 cm−1 and a reduction of the shoulder at 1,636 cm−1 with minimal changes in the region associated with turn/loop motifs. This suggests that incubation at an elevated temperature resulted in the reduction of weaker hydrogen bonds and the formation of stronger ones. Further incubation at an elevated temperature did not yield any notable changes. An opposite effect was observed for the Type 3 and Type 4 samples (Figs. 2C, 2D, 2G, 2H). Here, the main maximum positions shifted towards a larger wavenumber (from 1,623–1,624 to 1,625–1,626 cm−1). Surprisingly, the resulting FTIR spectra were almost completely identical to the one which formed when Type 1 fibrils were incubated (Figs. 2A, 2E). The Type 2 sample spectrum (Figs. 2B, 2F) experienced only very minor changes during incubation, which was to be expected, as it had a similar spectrum to the incubated fibril spectra, as seen for Type 1, 3 and 4 fibrils. Taken together, this indicates that the α-syn fibril polymorphism, which existed when aggregation occurred at 37 °C, was completely negated when the fibrils were incubated at an elevated temperature.

Figure 2 Fourier-transform infrared (FTIR) spectra and second derivatives of α-syn fibril samples before and after incubation at 60 °C and reseeding at 37 °C.

Type 1 (A, E), Type 2 (B, F), Type 3 (C, G) and Type 4 (D, H) fibril sample FTIR spectra and second derivatives. Black lines correspond to the control samples, orange–after 24 h of incubation at 60 °C, blue–after 48 h of incubation at 60 °C and green–48 h incubation samples reseeded at 37 °C. Dotted grey lines indicate the main maximum position of the initial sample FTIR spectrum. Superimposed FTIR spectra of all four fibril types before (I) and after 48 h of incubation at 60 °C (J).

In order to verify if this newly-formed fibril type is the result of restructurization or simply a temperature-induced association/clumping, each 48 h incubation sample was reseeded at 37 °C as described in the Materials and Methods section and the resulting aggregate FTIR spectra were scanned. Interestingly, all four of the α-syn samples retained the same identical spectra as were obtained after incubation at 60 °C. This suggests that the newly-formed fibril type is capable of templating its structure under lower temperature conditions as well and does not revert back to its original state. Comparing all four FTIR spectra before and after incubation shows how they all become practically identical and overlap with each other (Figs. 2I, 2J).

Considering that an elevated temperature resulted in the formation of aggregates with an identical secondary structure, it was interesting to compare them to fibrils which originally formed at 60 °C. Based on the kinetic data, alpha-synuclein aggregation proceeded significantly quicker when subjected to a higher temperature. The process average lag time was reduced three times (Fig. 3A) and the apparent rate constant of fibril elongation increased two times (Fig. 3B). The formed aggregate FTIR spectra were almost completely identical to the incubated fibril spectra, with the same main maximum position and overlapping spectra in the turn/loop motif region. This indicates that an elevated temperature leads to the same specific alpha-synuclein conformation, regardless if the aggregates were prepared at 37 °C or 60 °C.

Figure 3 Comparison of alpha-synuclein aggregation kinetics and resulting structure under different temperatures.

The lag time (A) and apparent rate constant of fibril elongation (B) at 37 °C and 60 °C (n = 8). Superimposed FTIR spectra of Type 1–4 incubated fibrils (blue color) and eight spectra of fibrils prepared at 60 °C (red color).

Since incubation led to changes in fibril secondary structure, it was important to also analyze any possible morphological differences. Comparing the AFM images of all three fibril Types (Figs. 4A–4D, with their 48 h incubation counterparts (Figs. 4E–4H)), it can be observed that both the initial and incubated samples contained long fibers with some of them having periodicity patterns. Based on a visual inspection, it may appear that the incubated samples contained a larger number of fibrils with periodicity patterns, however, such a conclusion is difficult to make based on AFM images alone.

Figure 4 Atomic force microscopy (AFM) images of different fibril types before and after incubation.

AFM images of Type 1 (A, E), Type 2 (B, F), Type 3 (C, G) and Type 4 (D, H) fibrils, as well as height (I), width (J) and periodicity (K) distribution before and after incubation respectively. All images are of identical 5 × 5 µm scale. Fibril height, width and periodicity were determined as described in the Materials and Methods section. Distribution box plots (n = 50) indicate the interquartile range and error bars are for one standard deviation.

Comparing the heights of fibrils (Fig. 4I) revealed that in all three cases, the average values were slightly higher for incubated samples (1–2 nm difference). This could be the result of the aforementioned morphological changes or fibril lateral association, however, the small difference in average values, coupled with no differences in fibril width (Fig. 4J) led to an assumption that incubation had no substantial effect on these parameters. Interestingly, the distances between periodic repeats of fibrils also did not change upon incubation (Fig. 4K), which means that the alterations in secondary structure did not have an influence on this morphological parameter. Additionally, all four cases (control and incubated), did not have a homogenous distribution, as both non-periodic and periodic fibrils could be detected in all samples. This suggests that the restructurization may not affect certain subgroups of aggregates, which may require a significantly longer time to change, as shown by Sidhu et al. (2017).

Since there was a slight increase in fibril height, there existed a possibility that parts of the unstructured protein region became incorporated into the aggregate core region. In order to examine this possibility, CD spectra of all four fibril types were scanned after each hour of incubation for 10 h, as were the 24 and 48 h samples. When comparing the initial and end-point samples, for Type 1 (Fig. 5A) and Type 3 fibrils (Fig. 5C) there was a slight increase in the fraction of beta-sheets and a reduction in either turn or unstructured regions. For the other two fibril types (Figs. 5B, 5D), there were no notable deviations observed. In all cases, however, the change was rather marginal and, due to the relatively large normalized root mean square deviation values (NRMSD) (Micsonai et al., 2015), could not be considered as statistically significant.

Figure 5 Alpha-synuclein fibril secondary structure element distribution during incubation at 60 °C.

The secondary structures for Type 1 (A), Type 2 (B), Type 3 (C) and Type 4 (D) fibrils were determined by scanning each sample’s CD spectra after different periods of incubation and fitting the data using BeStSel Protein circular dichroism spectra analysis software. The normalized root mean square deviation (NRMSD) of each sample’s secondary structure element distribution is displayed under their respective distribution graphs.

Examining the absorbance spectra of fibril-bound ThT (Figs. 6A–6D) revealed that there were no sizeable variations in either the maximum absorbance value position or the maximum absorbance itself. This indicates that the structural rearrangements did not cause significant changes to the amount of bound ThT molecules. Interestingly, the same cannot be said about the sample optical density. There was a clear disparity between the Type 2 fibril sample and the rest (Fig. 6E), where the Type 2 sample had an OD600 of 0.18, while for Type 1, 3 and 4 it was 0.10–0.12. After 24 h of incubation, there was only a minimal increase in OD600 for the Type 2 sample, while the other three aggregate samples experienced a substantial OD600 increase to 0.16–0.17. Additional incubation caused a further convergence of the OD600 values. Considering that the fibrils were resuspended into the initial reaction solution, which did not contain α-syn monomers, additional aggregation can be ruled out as a cause of such a significant increase in optical density. This leaves the possibility that aggregate structural rearrangement caused an increase in the sample light scattering properties. This is further supported by the fact that the Type 2 sample experienced the least significant change in OD600, similarly to it having the least notable FTIR spectrum shift.

Figure 6 Absorbance and fluorescence spectra of fibril-bound ThT during incubation at 60 °C.

Absorbance spectra of Type 1 (A), Type 2 (B), Type 3 (C) and Type 4 (D) fibril-bound ThT before and after incubation at 60 °C for 24 and 48 h. Optical density of all four samples before and after incubation (E), determined at 600 nm. Sample bound-ThT fluorescence EEM position changes over the course of 10 h of incubation (F). The change in EEM positions over time is represented as a color gradient in subfigure F (lighter color–shorter incubation time, darker color–longer incubation time). The initial 0 h sample EEM positions were measured after the samples reached 60 °C (the heating procedure was 10 min). Sample absorbance and fluorescence measurement procedures are described in the Materials and Methods section. Absorbance data is the average of three repeats.

In order to examine whether there was also a shift in bound-ThT EEM maximum positions, the four different fibril types were incubated as described in the Materials and Methods section at 60 °C and their EEM spectra were scanned every hour. The initial scans were performed after the samples reached 60 °C (after 10 min of incubation). During the first 4 h, the EEM positions shifted significantly (Fig. 6F), suggesting a change in the ThT-binding properties of the aggregates (as a note, the 60 °C EEM positions do not correlate to the Fig. 1 positions due to the elevated sample temperature, which reduces the dye’s fluorescence quantum yield and may influence the mode of binding). After 5–6 h of incubation, all four fibril type bound-ThT EEM positions converged to a similar position, which falls in line with the FTIR results. Comparing this information to the minimal change observed in ThT absorbance spectra, it appears that the structural alterations mainly affect the dye’s mode of binding, rather than the quantity of surface-associated molecules. These results suggest that most major changes to the fibril structures occur during the initial 4 h of incubation.

Discussion

Taking into consideration all the results presented in this work, it appears that different conformation α-syn fibrils undergo rapid restructurization at an elevated temperature. While physiological temperatures were also reported to cause aggregate maturation, the process spanned over several days or even months (Ma et al., 2013; Sidhu et al., 2017). In this case, based on changes in FTIR spectra and sample optical density, significant alterations to the secondary structure occurred in less than 24 h. If we use ThT fluorescence as an indicator of distinct structure aggregates, then the ThT excitation-emission matrix assay displays that most of the significant changes happened during the initial 4 h of incubation. If this is the case, then it is possible that α-syn fibril maturation is a condition-dependent process, which is greatly escalated at higher temperatures.

Another interesting aspect was the apparent convergence of multiple distinct conformation fibrils into one type of dominant aggregate with the increase of the solution’s temperature. The four fibril types with distinct FTIR spectra all rapidly changed into one specific conformation, which was also capable of replicating its structure at the initial 37 °C temperature, similarly to the initial four aggregate types. The structural alterations also do not appear to be caused by a significant increase in the fibril beta-sheet content (as seen during CD analysis), but rather due to a restructurization of existing secondary structure elements. This hints at a possible meta-stability of the fibrils generated at 37 °C and that their existence is only facilitated by a high energy barrier needed to cross into the higher stability structure. Such a hypothesis is supported by the FTIR spectra similarity of incubated fibrils and aggregates prepared at 60 °C (Fig. 3C). However, while this may be the case for pre-existing fibril conversion, the barrier does not appear to prevent the initial formation of such aggregates (Type 2 fibrils). In the AFM images, we can observe each initial sample containing fibrils with periodicity patterns. Such fibers seem to be the dominant type of structure after incubation and their presence in each sample suggests that they can readily form at 37 °C as well. Despite this appearance, it is difficult to quantitatively verify such an observation due to the nature of AFM imaging.

While this, in itself, is an interesting phenomenon, such a rapid aggregate restructurization may have a negative impact both during in vitro drug screenings and may even have certain physiological implications. If ThT fluorescence intensity or sample optical density are used as a means of identifying changes in aggregate concentration, then such restructurization events may provide false-negative or false-positive results, as seen in the OD600 (Fig. 6E) and ThT (Fig. 6F) assays. If experimental procedures involve the characterization of fibril structural parameters, then this type of rapid change to secondary structure may also lead to skewed experimental results and conclusions. Finally, if elevated temperatures can increase the incidence of the restructurization events, then this may be one of the many possible factors that influence the onset of neurodegenerative disorders, by stabilizing a certain fibril conformation.

Conclusions

Overall, it is apparent that an elevated temperature can not only significantly enhance the rate of alpha-synuclein fibril restructurization/maturation, but also result in the stabilization of a certain aggregate structure. This process also leads to changes in bound-ThT signal intensity and optical density, as well as alterations in fibril secondary structure, which should be taken into account during studies of both alpha-synuclein and other amyloid proteins.

Supplemental Information

Supplemental Information 1 Absorbance spectra and sample optical density raw data.

Click here for additional data file.

Supplemental Information 2 Circular dichroism spectra raw data.

Click here for additional data file.

Supplemental Information 3 Bound-ThT excitation-emission matrix raw data.

Click here for additional data file.

Supplemental Information 4 Fourier-transform infrared spectra and their second derivative raw data.

Click here for additional data file.

Supplemental Information 5 Alpha-synuclein aggregation at different temperatures raw data.

Click here for additional data file.

Supplemental Information 6 Atomic force microscopy images of fibril samples before and after 48 hours of incubation.

Click here for additional data file.

Additional Information and Declarations

Competing Interests

Author Contributions

Data Availability

The authors declare that they have no competing interests.

Mantas Ziaunys conceived and designed the experiments, performed the experiments, analyzed the data, prepared figures and/or tables, authored or reviewed drafts of the article, and approved the final draft.

Andrius Sakalauskas performed the experiments, authored or reviewed drafts of the article, and approved the final draft.

Kamile Mikalauskaite performed the experiments, authored or reviewed drafts of the article, and approved the final draft.

Vytautas Smirnovas conceived and designed the experiments, analyzed the data, authored or reviewed drafts of the article, and approved the final draft.

The following information was supplied regarding data availability:

The raw data is available in the Supplemental Files.

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
