# Peer review of "Rapid restructurization of conformationally-distinct alpha-synuclein amyloid fibrils at an elevated temperature"

_PeerJ, doi:10.7717/peerj.14137_

## Round 0.1 · original submission · Major Revisions

Although one of the reviewers recommended rejection, I have decided to give you an opportunity to revise your manuscript and address their concerns.

Reviewer 1 ·

Basic reporting

In this work, Ziaunys et al. report on the effects of elevated temperature in re-structurization of recombinant a-syn amyloid fibrils. Overall the manuscript is well-written, but there are several typos throughout the text that need to be corrected. Despite their effort, I find that the data reported by the authors are inconclusive and potentially indicate that the slight differences reported may emerge from technical differences rather that actually reported morphological differentiation, which I am not certain occurs even at the starting material. In my opinion, several experiments are required to provide more solid data supporting their claims and as a result, I can not recommend this manuscript for publication.

Experimental design

Method description requires improvement for clarity (see points below)
Research question is well-defined, not particularly original or filling a particular knowledge-gap.

Investigation was performed with several methods, that however fail to provide concrete results to support the claims and in certain cases provide conflicting results.

Validity of the findings

The results shown to not clearly substantiate the presence of different populations with this a-syn preparation. Using FTIR (or EEM for that matter) to tell apart subpopulations is not defining and at the very least requires a substantial amount of controlling with several independent preparations, as both are subjective to technical issues introducing variability.

Further to this, the additional experiments performed counter this argument indicating a single morphology that is actually retained between conditions.

Finally, the authors mention several cycles of sample manipulation, but assume the fibril content remains stable there, despite mentioning that it is conformationally variable even between a single condition.

Additional comments

Point 1. The point of providing raw data files is to facilitate reproducibility of the work. The raw files provided to not included in several cases a description of which experiment exactly they correspond to (which plot), or which sample each table is.

Point 2. How many samples were run in the first initial EEM experiment? Fig 1A shows circles of the same color overlapping, so it remains impossible to understand. Similarly, I can’t follow what the limits of each cluster type are and how they were defined exactly? How many replicates were included per type? Only the single circle/sample showed in the plot, no triplicates? What about random variability? The aggregation solution can differ slightly between preps in terms of buffer composition, monomer/oligomer/fibril/unordered aggregate fraction that could produce slight differences like the ones reported. More replicates would be necessary to exclude this. If more samples were obtained, how were they used for subsequent experiments? Were they pooled together in one sample?

Point 3. The authors use EEM plots to define different morphologies of a-syn fibrils, then use FTIR to joint them further together. According to Fig. 1, it seems that I-V that was joined together shows much more pronounced differences in the EEM compared to VI-VIII. Yet, the FTIR spectra clearly indicates the opposite. Is this not an indication that EEM might not be the best way to tell apart morphologies? There are several technical issues with EEM as a method (for instance IFE corrections) that support this further.

Point 4. Pooling type 1-3 samples. Based on the spectra provided (also the raw 2D data), I do not understand why was type VI isolated as a sample, but VII and VIII were pooled together? VIII has a beta-peak that is similar to VI rather than VII as the authors state in the manuscript, while all three sample differ significantly in the turn assignment anyways (its not minor, certainly compared to the beta-sheet differences reported).

Point 5. Importantly, to begin with, the fibrils were generated in 96-well plates (100uL) with a single bead and in the presence of ThT. These samples were then re-seeded in relatively unmatching conditions (in eppednorfs, with 2 beads, no extra ThT, cooling down – this I presume as it is not clearly mentioned) and then further supplemented with 2 more cycles of adding monomer. This fibrils were then used, for instance, for the FTIR measurement. But the conditions of fibrillation have changed during the intermittent steps, so how did the authors make sure they still had similar aggregates to the starting material? There is a plethora of work that indicates that multiple cycles (especially in altered conditions) can modify morphology of the “seed” aggregate. In this case, this is even more crucial as a point when, to begin with, the differences reported seem to be very small if there.

Point 6. FTIR after heating indicated a common structure resembling Type II fibrils. This structure remained even after re-seeding all samples at 37C. This comes as a strong indication to the previous point, that the conditions of the re-seeding are the dominant factor, not the temperature as suggested. AS FTIR measurement are sensitive to background subtraction and temperature control during measurements, minor differences such as the ones reported here need to be controlled with multiple measurements of parallel independent replicates. For that matter, how was FTIR measured (in solution, dried?). Was buffer subtraction performed?

Point 7. “Since incubation led to such significant changes in fibril secondary structure” I would not use such strong language in this case. The reported changes are not significant.

Point 8. AFM mapping indicating the formation of identical fibril morphologies. This re-inforces the aforementioned issues with EEM and FTIR to define subtypes. The reported differences are from a single preparation in each case, and could stem from technicalities during preparation rather than actually report differences.

Point 9. “EEM positions do not correlate to the Figure 1 positions due to different sample temperatures”. I do not follow this, isn’t the first point done at time point 0h? Is this not the same temperature? Should this not match?

Point 10. Have the authors considered the possibility of fragmentation occurring during the several re-seeding conditions? They use a glass bead in most incubation cycles which is notorious for this.

Point 11. “This indicates that the structural rearrangements did not cause significant changes to the amount of bound ThT molecules or their binding mode” Again, I am confused, if the authors report no changes to either the amount or binding mode of ThT on the fibril surface, how can they explain the differences of the EEM plots? Is this not then more probable due to slight mismatches of conditions?

Point 12. The authors use full length a-syn for these experiments. Cryo-EM has clearly documented that only a fraction of the protein forms the amyloid core of fibrils. Is it not possible that these slight alterations possible observed derived from the sequence not incorporated in the core? The so-called fuzzy coat? AFM, absorbance and slight FTIR changes could support this better.

Reviewer 2 ·

Basic reporting

The authors examined whether incubation at an elevated temperature can induce the restructurization of three different conformation alpha-synuclein amyloid fibrils. They concluded that three distinct amyloid fibrils transformed to fibrils with a common secondary structure and morphology at 60°C.

Experimental design

However, the experimental design and results are preliminary to validate the conclusion. More extensive experiments and discussion will be required.

Validity of the findings

1. They used ThT fluorescence, FTIR, and AFM to address the secondary structures and morphologies. They should also use CD spectroscopy to validate the structural changes they obtained with other methodologies.
2. They should try to form the amyloid fibrils at 60°C to confirm that they are similar to those obtained upon heating the distinct fibrils prepared at 37°C.
3. They should monitor the kinetics of conformational transition at 60°C in detail, which might be possible by using CD spectroscopy. In addition, this reviewer is interested in the temperature dependence of the formation of alpha-synuclein amyloid fibrils. Is the reaction at 60°C faster than that at 37°C?
4. Do the conformation of amyloid fibrils restructured at 60°C propagate at 37°C by seeding? The EEM position of merged fibrils at 60°C should be indicated in Fig. 1A.

---

## Round 0.2 · accepted · Accept

Although original reviewer was invited to evaluate your revised manuscript, I did not receive their comments and therefore I am making decision based on my own reading. In my view, all issues pointed by the reviewers were adequately addressed and the manuscript was amended accordingly. Therefore, the revised manuscript is acceptable now.